# Differential Effects of Multiple Dimensions of Poverty on Child Behavioral Problems: Results from the A-CHILD Study

**DOI:** 10.3390/ijerph182211821

**Published:** 2021-11-11

**Authors:** Yui Yamaoka, Aya Isumi, Satomi Doi, Manami Ochi, Takeo Fujiwara

**Affiliations:** 1Department of Global Health Promotion, Tokyo Medical and Dental University, Tokyo 113-8519, Japan; yamaoka.hlth@tmd.ac.jp (Y.Y.); isumi.hlth@tmd.ac.jp (A.I.); doi.hlth@tmd.ac.jp (S.D.); ochi.m.aa@niph.go.jp (M.O.); 2Japan Society for the Promotion of Science, Tokyo 102-0083, Japan; 3Department of Health and Welfare Services, National Institute of Public Health, Saitama 351-0197, Japan

**Keywords:** poverty, material deprivation, child behavioral problem, prosocial skill

## Abstract

The differential effects of low income and material deprivation—in particular, deprivation related to child educational needs—have not been well examined. This study aimed to examine the effects of low income and life-related and child-related deprivation on child behavioral problems. This study used data from first-grade students who participated in the Adachi Child Health Impact of Living Difficulty (A-CHILD) study in 2015, 2017, and 2019 (N = 12,367) in Japan. Material deprivation was divided into life-related deprivation (i.e., lack of items for a living) and child-related deprivation (i.e., lack of children’s books, etc.), and low income was assessed via annual household income. We assessed child behavioral problems and prosocial behavior using the Strengths and Difficulties Questionnaire. One in ten children belonged to low-income families, 15.4% of children experienced life-related deprivation, and 5.4% of children experienced child-related deprivation. While life- and child-related deprivation had significant adverse effects on behavioral problems, they had no association with prosocial behavior. The effects of low income were mediated by parental psychological distress (45.0% of the total effect) and the number of consulting sources (20.8%) on behavioral problems. The effects of life-related and child-related deprivation were mediated by parental psychological distress (29.2–35.0%) and the number of consulting sources (6.4–6.9%) on behavioral problems. Life-related and child-related deprivation, but not low income, are important for child mental health.

## 1. Introduction

Poverty is a major determinant of early experiences and environment, which play critical roles in child development [1]. A number of studies have reported pervasive adverse effects of poverty on children’s health [2], brain development [3,4], and educational outcomes [5]. A well-known measurement of poverty is low income, which represents the available financial resources in a household [6]. However, even in families with equivalent income, living environment and availability of necessities can differ based on how the families spend their money. Material deprivation is another indicator used to assess the lack of socially perceived necessities, items considered by the majority of the population to be essential for good standards of living [7]. A number of studies have defined material deprivation as a lack of standards of living, such as food insecurity, housing instability, and being behind on utility bills [8,9,10,11].

An important reason to assess poverty across multiple dimensions is that children in low-income families and/or experiencing material deprivation can have heterogeneous experiences. Although low income and material deprivation can overlap [12], living environments are diverse depending on how parents spend their time, material, and social resources with respect to their children. This constitutes the family investment model, which includes parental choices for standards of living, learning materials, and social stimulation for children [13]. Low income and material deprivation differentially affect multiple aspects of child development. For example, Gershoff et al. reported that income was associated with lower cognitive development, and material deprivation was associated with lower social–emotional competence [8]. A previous study that examined the effects of different types of hardships on child behavioral problems showed that difficulty paying bills and utility interruption were associated with externalizing behaviors, whereas difficulty paying bills, utility interruption, housing insecurity, and food hardship were associated with internalizing behaviors [9].

However, apart from the effects of low income and deprivation of living standards, little is known about how different types of deprivation relate to various aspects of children’s lives, such as how lacking educational resources or social activities affects child behavioral problems. UNICEF uses a child-right-based approach to define the seven dimensions of child poverty: nutrition, clothing, educational resources, leisure activities, social activities, information access, and quality of housing [14]. Therefore, there is a need to understand the independent effects of deprivation of children’s needs to provide specific support for children and their families.

An understanding of the mechanisms governing the effects of low income and deprivation on child behavioral outcomes would aid the development of specific support strategies for families and children. The effects of low income and deprivation on child outcomes are often explained by two models: the family investment model (i.e., economic hardship affects how families invest their money and resources) and family stress model (i.e., economic hardship affects parental psychological distress and parenting practices) [8,9,10,15]. Based on the family stress model of economic hardship [16], increased economic pressure is associated with parenting stress and parenting behaviors, which are indirectly associated with child behavioral problems. Thus, providing parents with psychological support may be necessary to mitigate the adverse effects of low income or deprivation on child behavioral outcomes. In addition, prior studies have reported that people living in poverty are more likely to become socially isolated [17,18,19] and have fewer resources to access [20]. The buffering effects of social support on social stressors have been demonstrated [21]. Further, greater parental social support is significantly associated with lower child behavioral problems [22,23]. Therefore, providing parents with social support can be another strategy for preventing adverse child behavioral outcomes.

This study aimed to examine (1) the prevalence of combinations in three dimensions of poverty: low income, deprivation of children’s needs, and deprivation of living standards among Japanese school-aged children, (2) whether deprivation of children’s needs has a different deteriorating effect on child behavioral problems compared to low income or life-related deprivation, and (3) the mediating effects of parental psychological distress or social support related to low income and deprivation on child behavioral outcomes.

## 2. Materials and Methods

### 2.1. Study Samples

This study used data from Adachi Child Health Impact of Living Difficulty (A-CHILD), which aimed to examine the circumstances of children living in poverty and their health status in all 69 primary schools located in Adachi City [24]. Adachi City has a population of approximately 690,000 and is one of the 23 special wards in Tokyo Prefecture. This study focused on first-grade students who participated in the A-CHILD surveys conducted in 2015, 2017 and 2019. The items examined in the questionnaires included the demographics of the households, parents, and children, parenting behaviors, the psycho-social circumstances of the children and parents, and the lifestyle and behaviors of the children. The questionnaires were completed by the parents of the target children and returned through the child’s school in an anonymous sealed envelope. Figure 1 shows the characteristics of the study subjects in each survey (response rate: 78.8–81.6%). Overall, 12,367 subjects were included in the analysis after excluding children with missing outcome variables.

### 2.2. Low Income and Material Deprivation

Parents of first-grade students responded to a question inquiring about their total household income for the past year. The response was categorized as <3.0, 3.0 to <6.0, 6.0 to <10.0, or ≥10.0 million yen (1.0 million yen ≒ 9500 USD). We defined low income as income below 3.0 million yen (≒2800 USD) based on the definition of households in living difficulty used in a previous study [24]. We created a separate category for a non-response to the income question (12.7%). The question about household income was asked after items on demographics, the health conditions of the parents and children, and the lifestyle of the children.

We asked the parents about material deprivation and payment difficulties after the items on household income using two questions. The first question asked parents to select, from the following list, all the materials they lacked at home for financial reasons: (a) age-appropriate picture books and books for the child, (b) sports equipment, stuffed animals, and toys for the child, (c) a place to do homework for the child, (d) a washing machine, (e) a rice cooker, (f) a vacuum cleaner, (g) heating equipment, (h) cooling equipment, (i) a microwave, (j) a phone, (k) a bath, (l) a sufficient number of beds or futons for household members, (m) savings of more than 50,000 yen (≒474 USD) for unexpected expenses, and (n) none of above. The second question asked parents to select, from the following list, the items they were unable to make payments for over the last year: (o) fees for school trips/excursions, (p) transportation or participation fees for extracurricular school activities, (q) school lunch fees, (r) rent, (s) mortgage repayments, (t) electricity bills, (u) gas bills, (v) water bills, (w) phone bills, (x) premiums for public pension, health insurance, and long-term care insurance, (y) transportation, and (z) none of above. If parents selected at least one of the six items from a–c and o–q, their children were defined as experiencing child-related deprivation. If parents selected at least one of the 18 items from items d–m and r–y, their children were defined as experiencing life-related deprivation.

### 2.3. Child Behavioral Problems

We used the validated Japanese version of the Strength and Difficulties Questionnaire (SDQ) to assess behavioral problems [25]. Participants completed the SDQ after answering questions on material deprivation, parenting behaviors, parental psychological distress, and the number of consulting sources. Parents provided responses to 25 items on their child’s behavior during the past six months using a three-item scale (2 = very true, 1 = true, and 0 = not true). The SDQ was composed of five subscales with five items each (range: 0–10): emotional problems, conduct problems, hyperactive/inattention problems, peer relationship problems, and prosocial behavior. The total SDQ score was obtained by summing the scores from four subscales, except for prosocial behavior (range: 0–40). Higher SDQ total and subscale scores, except for prosocial behavior, indicate greater problematic behaviors in a child. In contrast, a higher prosocial behavior score indicates that the child has more favorable social behaviors.

### 2.4. Covariates

We also assessed the following demographic variables: child’s age in months and sex, number of parents in the household (two, one, or none), number of grandparents in the household (none, one, two or more), number of children in the household (one, two, three, four or more), maternal age (<30, 31–34, 35–39, 40–44, ≥45 years old), maternal education (high school or less, technical school or junior college, university or higher), and employment status (full/part-time or not working). Maternal mental health status was assessed using the Japanese version of Kessler 6 (K6). K6 assesses the frequency of six psychological distress symptoms over the last month (0 = none of the time, 4 = all the time). A score of 5 or over is a widely used cut-off for screening mood and anxiety disorders [26]. Parents were also asked who they consulted when they had troubles and worries. Parents responded to the question by choosing according to the following list: spouse/partner, your parents, parents of spouse/partner, your siblings or relatives, friends or acquaintances living in the neighborhood, friends or acquaintances not living in the neighborhood, colleagues, teachers or school counselors, public agency or welfare agency, private or telephone counseling service, physicians or nurses, websites, and others. The degree of social support was assessed according to the number of consulting sources indicated by each parent. The number of consulting sources was categorized from none, 1–2, 3–4, to 5+ sources.

### 2.5. Statistical Analysis

First, we identified the demographic characteristics of the children and their families and the proportion with low income and deprivation in each survey. Second, multivariate linear regression analyses were performed to examine the relationships between the income, deprivation, and SDQ total score and the prosocial score. We added demographic variables, year of survey, and low income to model 1. In addition to low income, we further added child-related deprivation to model 2 and life-related deprivation to model 3. Finally, we examined the effects of mediators (i.e., parental mental health and low social support) between low income and deprivation and child behavioral problems. We used structural equation modeling (SEM) to examine the indirect effects of three types of poverty on SDQ total scores via parental psychological distress and the number of consulting sources. Outcome variables were standardized to compare effects by income and deprivation (mean = 0, SD = 1.0). The analyses were conducted using Stata/MP version 16.1 (STATA Corp., College Station, TX, USA).

## 3. Results

Table 1 shows the demographic characteristics of the children and their households who participated in the surveys in each year. In Adachi City, most households were nuclear, consisting of two parents and two children. Less than one in ten families comprised single-parent households and lived with one or more grandparents. One-tenth of the mothers were aged 45 years or older, and the proportion with higher maternal age increased over the three survey years (45 years old and over: 9.7% to 12.0%). One-quarter of the mothers had graduated from university or higher, with the proportion increasing (20.4% to 29.0%) over five years. One-third of the mothers experienced psychological distress (K6 score: 5 and over). Less than 10% of parents did not have anyone to consult when they had troubles and worries.

Table 2 shows the proportion of first-grade children with low income and child-related and life-related deprivation. Overall, 9.8% of children belonged to low-income families, 5.4% experienced some type of child-related deprivation, and 15.2% experienced some type of life-related deprivation. The most frequent source of deprivation among child-related items was a place to study (2.9%), and among life-related items, it was saving for unexpected expenses (10.5%). Approximately two-thirds of the children did not experience any dimensions of poverty, while 1.4% of children experienced all three dimensions of poverty (Appendix A).

Table 3 shows the relationship between behavioral problems and each dimension of poverty. Low income was not significantly associated with the SDQ total score (β = 0.057). In contrast, child-related deprivation and life-related deprivation were significantly associated with higher behavioral difficulty (child: β = 0.252, *p* < 0.001; life: β = 0.230, *p* < 0.001). After controlling for low income and deprivation, parental psychological distress and fewer consulting sources (i.e., low social support) were significantly associated with higher behavioral difficulty (psychological distress: β = 0.524–0.544, *p* < 0.001; no source to consult: β = 0.069–0.075, *p* < 0.001).

Table 4 shows the relationship between prosocial behavior and each dimension of poverty. Prosocial skill was not associated with any of the three dimensions of poverty examined in models 1 to 4. After controlling for low income and deprivation, parental psychological distress and fewer consulting sources (i.e., low social support) were significantly associated with lower prosocial behavior (psychological distress: β = −0.109–0.110, *p* < 0.001; no source to consult: β = −0.408–0.409, *p* < 0.001).

Table 5 shows the results of the analysis of the mediating effects of parental psychological distress and the number of consulting sources associated with low income, life-related or child-related deprivation, and child behavioral problems. Total effects are the sum of direct effects and two indirect effects (parental psychological distress or the number of consulting sources). The direct effects in SEM are shown in Appendix A. Both direct and indirect effects were divided by the total effects to determine the proportional contribution of each effect. Indirect effects of low income were mediated by parental psychological distress (45.0% of total effect) and the number of consulting sources (20.8%) on the SDQ total score. Similarly, the indirect effects of life-related and child-related deprivation were mediated by psychological distress (life-related: 29.2%; child-related: 35.0%) and the number of consulting sources (life-related: 6.4%; child-related: 6.9%).

## 4. Discussion

This study examined the effects of the co-occurrence of multiple dimensions of poverty among Japanese children. More than one in five children experienced low income, or life-related deprivation, or child-related deprivation, and 1.4% of children experienced all three. We found independent associations of low income and child- and life-related deprivation with multiple aspects of child behavioral development apart from low income. The effects of low income and life-related and child-related deprivation on behavioral problems were mediated by parental psychological distress and low social support.

### 4.1. Prevalence of Multidimensional Poverty

Life-related deprivation was the most common dimension of poverty present among our study sample. Among our life-related deprivation items, a lack of savings for unexpected expenses was the most prevalent (10.5%), followed by payment for premiums (5.7%), which would increase financial distress among parents. Subjective financial stress is commonly experienced among families both with and without income poverty, and the combination of financial stress and material deprivation has greater adverse effects on child behavioral outcomes [10]. Therefore, assessing financial stability is important for evaluating poverty and deprivation.

The prevalence of child-related deprivation was lower (5.4%) than the prevalence of life-related deprivation. For example, the most selected child-related deprivation item in this study was a lack of a place to study (2.9%), followed by a lack of books (1.8%). Because the study subjects were first-grade children, their need for educational resources and opportunities to participate in social/school activities will continue to increase as they grow up. Some of the disadvantages experienced by children living in poverty in their school lives include struggling with transportation fees, school clothing and textbook costs, choices of subjects that may require expensive materials, and missing school trips [27]. It is essential to protect children’s right to receive an education and to promote social–emotional, cognitive, and academic development among all children, irrespective of whether they live in poverty.

Approximately one in ten children came from low-income families (<3.0 million per year), although 12.7% of participants did not respond to the income item in our study. In 2016, the child poverty rate, the proportion of children in families with an income less than 50% of the median equivalent disposable income, was 13.9% in Japan, which is just slightly higher than the OECD average (13.1%) [28]. However, the differences in the prevalence of child-related deprivation between families with and without low income were smaller (19% and 4%) than the differences in the prevalence of life-related deprivation between families with and without low income (43% and 11%). This finding suggests that parents tried to provide the essential materials and pay the necessary costs for children regardless of income status. As indicated in previous studies [8,10], income status does not capture the whole picture of economic hardship. Further continuous research is needed to evaluate the effects of multiple dimensions of poverty on children’s lives.

### 4.2. Direct Effects of Life-Related and Child-Related Deprivation

We found significant direct effects of child- and life-related deprivation on behavioral problems, while low income was not significantly associated with behavioral outcomes. These results are consistent with those of a previous study that also reported that poor child outcomes are associated with material deprivation, but not with low income [9]. This indicates that material resources have independent effects on the daily lives and needs of children that are not represented by income. Moreover, we found that prosocial behavior was not associated with low income or child-/life-related deprivation, which is consistent with a previous finding on behavioral problems among children aged 5 years old [9]. Prosocial behaviors, such as sharing, helping, and showing empathy, develop from early childhood through biological factors, socialization experiences, and psychological processes [29]. For example, children learn prosocial behaviors from their parents’ behaviors, such as caring, role-modeling, verbal encouragement, and praise [30]. Single parenthood, family structure transitions, and poverty in childhood have been shown to result in fewer prosocial behaviors in adulthood [30,31], which are not in line with our findings. Our study subjects, children in first grade, are still developing prosocial skills and will continue to do so through middle childhood into adolescence as they undergo cognitive and empathic development [32]. Further research is needed to examine whether low income and deprivation of living materials and children’s needs have adverse effects on prosocial development in later adolescence through adulthood.

### 4.3. Mediating Effects of Parental Psychological Distress and Social Support

This study confirmed that parental psychological distress mediated associations between poverty and child behavioral problems. Yeung et al. [14] reported that the association between family income and externalizing behavior was mediated by maternal emotional distress and parenting practices. Gershoff et al. [8] also reported that material deprivation was related to increased parental stress in all income quintile groups, and that parental stress was associated with parent investment and parenting behavior. Although low income did not have a direct effect on child outcome, its indirect effect via psychological distress constituted over 40% of the total effects. Child-related and life-related deprivation had both direct and indirect effects. Thus, providing parents with psychological support is necessary to improve behavioral problems among children living with low income and deprivation.

Social support also mediated associations between poverty and child behavioral problems. A prior study reported that maternal-perceived social support was associated with lower behavioral problems and higher prosocial behavior [33]. The present study confirmed that low income and deprivation were linked to lower social support, which can subsequently lead to higher behavioral problems. Even though low income did not have direct effects on child outcomes, it had an indirect effect via social support that comprised 20% of the total effects. Child-related and life-related deprivation had larger direct effects (approximately 60%) than indirect effects (6% of the total effects). Thus, providing parents with a source of social support may be an additional strategy to improve behavioral problems among children living with low income and deprivation.

### 4.4. Limitations

There are several limitations in this study. First, because of the cross-sectional study design, whether or not there is a causal relationship between low income or deprivation and children’s mental health remains unclear. Some parents may not provide their children with the necessary items because their children are exhibiting problematic behaviors. Longitudinal research is needed to understand the association between low income and deprivation and child behavioral outcomes. Second, assessment of low income or deprivation may have been biased due to subjective reporting. Future research should use objective measures of low income and deprivation, such as tax payment records. Third, the order of items in the questionnaire may have affected parents’ responses about their child’s behavior, as parents answered questions about poverty and material deprivation before completing the SDQ. We placed the SDQ at the end of questionnaire based on consideration of the burden of answering the 25 questions in the SDQ. Further research using clinical assessments of child behavioral problems to examine the association between child mental health and multiple dimensions of poverty is needed to confirm our findings. Finally, some child-related deprivation items may not be considered necessary by parents if their children do not desire them. Deprivation based on what a child wants is strongly associated with child subjective well-being [34]. Child-centric measurement is also important for improving children’s quality of life.

## 5. Conclusions

Japanese school-aged children with multiple dimensions of poverty have heterogeneous experiences. Deprivation of items related to daily living and children’s needs was independently associated with behavioral problems. In addition to providing material support for living and children’s needs, it is also essential to provide psychological and social support to parents living in poverty to prevent child behavioral problems.

## Figures and Tables

**Figure 1 ijerph-18-11821-f001:**
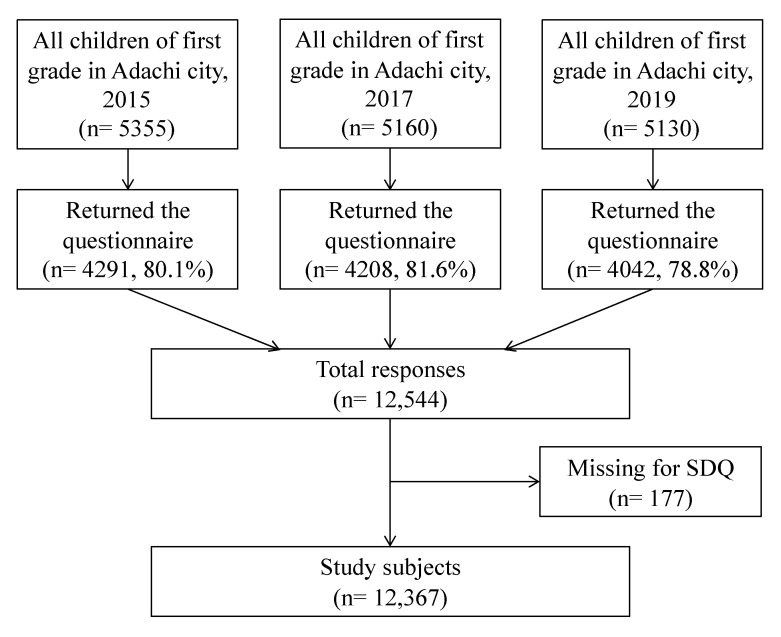
Flowchart of the study subjects.

**Table 1 ijerph-18-11821-t001:** Demographics of participants in 2015, 2017, and 2019.

		Total		2015		2017		2019	
		(n = 12,367)		(n = 4219)		(n = 4168)		(n = 3980)	
		Mean/N	S.D./%	Mean	S.D.	Mean	S.D.	Mean	S.D.
Child age	(Months old)	84.7	3.5	85.2	3.56	84.5	3.42	84.4	3.52
		n	%	n	%	n	%	n	%
Child sex	Female	6073	49.1%	2048	48.5%	2055	49.3%	1970	49.5%
	Male	6286	50.8%	2166	51.3%	2110	50.6%	2010	50.5%
	Missing	8	0.1%	5	0.1%	3	0.1%	0	0.0%
Number of parents living at home	Two parents	11,144	90.1%	3785	89.7%	3765	90.3%	3594	90.3%
	One parent	1114	9.0%	393	9.3%	370	8.9%	351	8.8%
	No parents	109	0.9%	41	1.0%	33	0.8%	35	0.9%
	Missing	0	0.0%	0	0.0%	0	0.0%	0	0.0%
Number of grandparents living at home	No grandparents	11,127	90.0%	3765	89.2%	3758	90.2%	3604	90.6%
	One grandparent	677	5.5%	265	6.3%	227	5.4%	185	4.6%
	Two+ grandparents	563	4.6%	189	4.5%	183	4.4%	191	4.8%
	Missing	0	0.0%	0	0.0%	0	0.0%	0	0.0%
Number of children in the household	One child	2539	20.5%	871	20.6%	854	20.5%	814	20.5%
	Two children	6351	51.4%	2167	51.4%	2107	50.6%	2077	52.2%
	Three children	2797	22.6%	948	22.5%	974	23.4%	875	22.0%
	Four+	676	5.5%	229	5.4%	233	5.6%	214	5.4%
	Missing	4	0.0%	4	0.1%	0	0.0%	0	0.0%
Maternal age	<30	661	5.3%	218	5.2%	244	5.9%	199	5.0%
	30–34	2231	18.0%	756	17.9%	753	18.1%	722	18.1%
	35–39	4104	33.2%	1444	34.2%	1372	32.9%	1288	32.4%
	40–44	3765	30.4%	1301	30.8%	1277	30.6%	1187	29.8%
	45+	1314	10.6%	411	9.7%	425	10.2%	478	12.0%
	Missing	292	2.4%	89	2.1%	97	2.3%	106	2.7%
Maternal education	High school	4064	32.9%	1507	35.7%	1361	32.7%	1196	30.1%
	Technical school/junior college	4922	39.8%	1750	41.5%	1673	40.1%	1499	37.7%
	University	3017	24.4%	862	20.4%	999	24.0%	1156	29.0%
	Other/unknown	364	2.9%	100	2.4%	135	3.2%	129	3.2%
Maternal employment	Yes	8592	69.5%	2766	65.6%	2884	69.2%	2942	73.9%
	No	3775	30.5%	1453	34.4%	1284	30.8%	1038	26.1%
	Missing	0	0.0%	0	0.0%	0	0.0%	0	0.0%
Maternal mental condition	K6: <5	8522	68.9%	2,998	71.1%	2872	68.9%	2652	66.6%
	K6: 5+	3794	30.7%	1204	28.5%	1284	30.8%	1306	32.8%
	Missing	51	0.4%	17	0.4%	12	0.3%	22	0.6%
Number of consulting sources	None	1174	9.5%	474	11.2%	350	8.4%	350	8.8%
	One to two	7848	63.5%	2,565	60.8%	2704	64.9%	2579	64.8%
	Three to four	2964	24.0%	1022	24.2%	993	23.8%	949	23.8%
	Five+	371	3.0%	148	3.5%	121	2.9%	102	2.6%
Outcome variable		Mean	S.D.	Mean	S.D.	Mean	S.D.	Mean	S.D.
SDQ total	(Range: 0–40)	10.3	5.41	9.93	5.31	10.4	5.42	10.5	5.46
SDQ-prosocial	(Range: 0–10)	6.65	2.07	6.61	2.04	6.69	2.06	6.66	2.09

**Table 2 ijerph-18-11821-t002:** Proportion with low income and child-related and life-related deprivation.

			N	%
Income	≥3,000,000 yen		9582	77.5%
	<3,000,000 yen		1210	9.78%
	No response		1575	12.7%
Child-related deprivation	Material hardship	Books	217	1.75%
		Toys	143	1.16%
		Place to study	364	2.94%
	Payment difficulty	Trips	78	0.63%
		Excursions	36	0.29%
		School lunch	162	1.31%
	Any child-related deprivation		672	5.43%
Life-related deprivation	Material hardship	Washing machine	31	0.25%
		Cooking appliances	27	0.22%
		Cleaning appliances	33	0.27%
		Heating	54	0.44%
		Air conditioning	54	0.44%
		Oven	41	0.33%
		Phone	147	1.19%
		Bath	34	0.27%
		Bed	315	2.55%
		Savings	1304	10.5%
	Payment difficulty	Rent	197	1.59%
		Mortgage	81	0.65%
		Electricity	218	1.76%
		Gas	194	1.57%
		Water	189	1.53%
		Phone	207	1.67%
		Premiums	704	5.69%
		Transportation	45	0.36%
	Any life-related deprivation		1904	15.2%
Combinations of dimensions of poverty				
None of the three dimensions of poverty			8385	67.8%
Low income and child-related deprivation			53	0.43%
Low income and life-related deprivation			340	2.75%
Child-related and life-related deprivation			244	1.97%
All three dimensions of poverty			177	1.43%

**Table 3 ijerph-18-11821-t003:** Relationship between multiple types of poverty and children’s behavioral outcomes (SDQ total score).

		Model 1			Model 2			Model 3		
		Coef	S.E.	*p*-Value	Coef	S.E.	*p*-Value	Coef	S.E.	*p*-Value
Child sex (ref: female)	Male	0.238	0.017	<0.001	0.236	0.017	<0.001	0.237	0.017	<0.001
Child age	(Months)	−0.028	0.002	<0.001	−0.028	0.002	<0.001	−0.028	0.002	<0.001
Number of parents living at home (ref: two parents)	One parent	0.033	0.037	0.374	0.020	0.037	0.590	0.017	0.037	0.647
	None, other	0.365	0.109	0.001	0.353	0.109	0.001	0.361	0.109	0.001
Number of grandparents living at home (ref: none)	One grandparent	0.107	0.039	0.006	0.109	0.039	0.005	0.107	0.039	0.006
	Two+ grandparents	0.043	0.043	0.319	0.048	0.043	0.259	0.046	0.043	0.283
Number of children in the household (ref: one child)	Two children	−0.189	0.023	<0.001	−0.188	0.023	<0.001	−0.190	0.023	<0.001
	Three children	−0.259	0.027	<0.001	−0.265	0.027	<0.001	−0.274	0.027	<0.001
	Four+ children	−0.232	0.042	<0.001	−0.256	0.042	<0.001	−0.274	0.043	<0.001
Maternal age	30–34 (years old)	−0.088	0.043	0.040	−0.085	0.043	0.046	−0.073	0.043	0.088
(ref: age < 30)	35–39	−0.181	0.041	<0.001	−0.177	0.041	<0.001	−0.161	0.041	<0.001
	40–44	−0.202	0.041	<0.001	−0.196	0.041	<0.001	−0.175	0.041	<0.001
	45+	−0.266	0.047	<0.001	−0.262	0.046	<0.001	−0.242	0.046	<0.001
Maternal education	High school or less	0.185	0.024	<0.001	0.176	0.024	<0.001	0.159	0.024	<0.001
(ref: university/grad school)	Technical school, small college	0.060	0.022	0.007	0.057	0.022	0.010	0.050	0.022	0.025
	Other	0.128	0.065	0.047	0.122	0.065	0.058	0.108	0.064	0.093
Maternal unemployment	(ref: employed)	−0.060	0.019	0.001	−0.057	0.019	0.002	−0.053	0.019	0.005
Parental psychological distress (ref: no)	Yes (K6: 5+)	0.544	0.019	<0.001	0.530	0.019	<0.001	0.524	0.019	<0.001
Number of consulting sources (ref: five+)	Three to four	0.039	0.053	0.462	0.039	0.053	0.460	0.043	0.053	0.416
	One to two	0.146	0.051	0.004	0.145	0.051	0.004	0.144	0.051	0.004
	None	0.319	0.058	<0.001	0.303	0.058	<0.001	0.301	0.058	<0.001
Year of survey (ref: 2015)	2017	0.053	0.021	0.012	0.057	0.021	0.007	0.057	0.021	0.007
	2019	0.069	0.022	0.001	0.071	0.022	0.001	0.075	0.022	0.001
Low income	<3M yen	0.057	0.034	0.095	0.033	0.034	0.340	0.013	0.034	0.695
(ref: >3M yen)	Missing	0.013	0.026	0.628	0.011	0.026	0.678	0.012	0.026	0.654
Child deprivation (ref: none)	One or more				0.252	0.040	<0.001			
Life deprivation (ref: none)	One or more							0.220	0.026	<0.001

Coefficients are presented in SD units.

**Table 4 ijerph-18-11821-t004:** Relationship between multiple types of poverty and children’s prosocial behavior (SDQ prosocial).

		Model 1			Model 2			Model 3		
		Coef	S.E.	*p*-Value	Coef	S.E.	*p*-Value	Coef	S.E.	*p*-Value
Child sex (ref: female)	Male	−0.355	0.018	<0.001	−0.355	0.018	<0.001	−0.355	0.018	<0.001
Child age	(Months)	0.019	0.003	<0.001	0.019	0.003	<0.001	0.019	0.003	<0.001
Number of parents living at home (ref: two parents)	One parent	0.072	0.039	0.063	0.072	0.039	0.066	0.071	0.039	0.067
	None, other	−0.172	0.114	0.132	−0.173	0.114	0.131	−0.172	0.114	0.132
Number of grandparents living at home (ref: none)	One grandparent	−0.036	0.041	0.375	−0.036	0.041	0.377	−0.036	0.041	0.376
	Two + grandparents	0.045	0.045	0.319	0.045	0.045	0.315	0.045	0.045	0.316
Number of children in the household (ref: one child)	Two children	−0.027	0.024	0.269	−0.027	0.024	0.271	−0.027	0.024	0.269
	Three children	−0.004	0.028	0.887	−0.004	0.028	0.877	−0.005	0.028	0.862
	Four + children	0.020	0.044	0.649	0.019	0.044	0.674	0.018	0.045	0.692
Maternal age	30–34 (years old)	−0.122	0.045	0.007	−0.121	0.045	0.007	−0.121	0.045	0.007
(ref: age < 30)	35–39	−0.160	0.043	<0.001	−0.160	0.043	<0.001	−0.159	0.043	<0.001
	40–44	−0.251	0.043	<0.001	−0.251	0.043	<0.001	−0.249	0.043	<0.001
	45+	−0.301	0.049	<0.001	−0.301	0.049	<0.001	−0.299	0.049	<0.001
Maternal education	High school or less	0.020	0.025	0.428	0.019	0.025	0.441	0.018	0.025	0.469
(ref: university/grad school)	Technical school, small college	0.002	0.023	0.941	0.002	0.023	0.947	0.001	0.023	0.963
	Other	0.051	0.068	0.450	0.051	0.068	0.454	0.050	0.068	0.462
Maternal unemployment	(ref: employed)	0.051	0.020	0.010	0.051	0.020	0.010	0.051	0.020	0.010
Parental psychological distress (ref: no)	Yes (K6: 5+)	−0.109	0.020	<0.001	−0.109	0.020	<0.001	−0.110	0.020	<0.001
Number of consulting sources(ref: five+)	Three to four	−0.151	0.055	0.006	−0.151	0.055	0.006	−0.151	0.055	0.006
	One to two	−0.284	0.053	<0.001	−0.284	0.053	<0.001	−0.284	0.053	<0.001
	None	−0.408	0.061	<0.001	−0.409	0.061	<0.001	−0.409	0.061	<0.001
Year of survey (ref: 2015)	2017	0.062	0.022	0.005	0.062	0.022	0.005	0.062	0.022	0.005
	2019	0.053	0.023	0.019	0.053	0.023	0.019	0.053	0.023	0.019
Low income	<3M yen	0.040	0.036	0.259	0.039	0.036	0.280	0.038	0.036	0.297
(ref: >3M yen)	missing	0.022	0.028	0.435	0.021	0.028	0.438	0.022	0.028	0.437
Child deprivation (ref: none)	One or more				0.015	0.042	0.721			
Life deprivation (ref: none)	One or more							0.013	0.027	0.621

Coefficients are presented in SD units.

**Table 5 ijerph-18-11821-t005:** Analysis of mediators of child behavioral difficulties (SDQ total).

	Income			% of total effect
	coef	S.E.	*p*-value	
Total effect	0.129	0.035	<0.001	
Direct effect	0.060	0.034	0.077	46.3%
Indirect effect via parental psychological distress	0.058	0.008	<0.001	45.0%
Indirect effect via number of consulting sources	0.027	0.004	<0.001	20.8%
	Life-related deprivation			% of total effect
	coef	S.E.	*p*-value	
Total effect	0.355	0.026	<0.001	
Direct effect	0.234	0.025	<0.001	66.0%
Indirect effect via parental psychological distress	0.104	0.007	<0.001	29.2%
Indirect effect via number of consulting sources	0.023	0.003	<0.001	6.4%
	Child-related deprivation			% of total effect
	coef	S.E.	*p*-value	
Total effect	0.458	0.041	<0.001	
Direct effect	0.274	0.040	<0.001	59.9%
Indirect effect via parental psychological distress	0.160	0.012	<0.001	35.0%
Indirect effect via number of consulting sources	0.031	0.005	<0.001	6.9%

## Data Availability

Data sharing is not applicable to this article.

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
