# Peer review of "Differential Effects of Multiple Dimensions of Poverty on Child Behavioral Problems: Results from the A-CHILD Study"

_ijerph, 2021, doi:10.3390/ijerph182211821_

Round 1

Reviewer 1 Report

I reckon that the authors addressed all the points raised by both reviewers and I found the revised version of the manuscript ready for publication.

Reviewer 2 Report

Authors meet the said modification to improve the quality of manuscript.

This manuscript is a resubmission of an earlier submission. The following is a list of the peer review reports and author responses from that submission.

Round 1

Reviewer 1 Report

Thank you for giving me the opportunity to review your paper Differential effects of low income and life-related and child-related deprivation on child behavioral problems: Results from A-CHILD Study.You clearly spent time and resources collecting your data. Here are my observations on which manuscript in its current form can be improved. Wish you best of luck for your research. 

1. Some statements in the introduction have been claimed without supporting references. Add some more relevant references supporting your statements.It will be better you add one or two paragraph in  introduction section to explain your research work vision.

2.From my point of view, the literature section must be add which will improve and strengthen your manuscript. Authors need to cite more latest papers in the relevant field to build an up-to-date picture of work. Add recent articles from the journal as well.

3.The methodology section must be provide more appropriate details of population, sample, sampling technique and data collection procedure. this section need refining and rephrasing.

4.In the analysis and results section is need to be improved by adding more clarification of it.

5.The conclusion needs to be clearer about how to establish a virtuous circle of low income , life-related and child-related deprivation on child behavioral problems can be overcome.

6. The title must not used same conjunction words multiple time.

7.check the English language constructions through as there is often confusion of terms or misleading statements.

Reviewer 2 Report

The present manuscript reports results from a cross-sectional study conducted on large samples of parents of first-grade students across three different years. The authors aimed to examine the differential effects of low income, material deprivation (subdivided into life-related and child-related deprivation) on child mental health (more specifically problematic behaviors). They present correlational data show that material deprivation is more associated with behavioral problems than low income.

Given the multidimensional nature of poverty and the complexity to get a full picture of the dynamics by which poverty impacts children, the topic is of high-interest. The manuscript is clearly written and easy to follow.

That being said, I have several concerns about the present research.

First, the Methods section is incomplete and makes it impossible for the reviewer to assess the procedure. It seems that the research was not pre-registered, which is regrettable for different reasons including the fact that further information on the Methods can usually be found in pre-registration forms. More specifically, I would have liked to know the presentation order of the different measures. Since nothing is said about that, I assumed that the presentation order was not counterbalanced and that respondents took the different measures in the same order as they are presented in the manuscript. If so, this may be problematic. Indeed, if parents first started by giving information about their income, following by the measures of material deprivation right before assessing their children behaviors, there is a risk that the first measures triggered a form of confirmation bias among poor respondents. IHaving to think about their precarious situation may have led them to perceive more problematic behaviors from their children in reason of a naive and stereotypic belief that “the poors do not behave well”. Since I am completely inferring the procedure and the presentation order of the measures, I may be completely wrong. In any case, further information should be provided in the Methods section.

Secondly, I found frustrating that the authors did not offer a more complex analysis of their data. It seems that their aim was merely to assess to what extent different dimensions of poverty are related to child’s behavioral problems independently from other potential factors that they also measured. But looking at the results presented in Table 3, the reader can see that there are numerous other significant correlations. Among them, the strong correlations of Maternal psychological distress with SDQ total score in all 4 models is compelling and echoes previous results in the literature. But the authors neither comment on those additional correlations nor provide further analyses that could give a clearer picture of the relations between the different variables of interest (income, child and material deprivation) and the other variables such as maternal psychological distress that seems to be even more strongly related to children’ behavioral problems. At the end, it leaves us with a set of independent correlations that do not say much about the complex interplay of different factors involved in the experience of poverty (income, material and child deprivation, maternal psychological distress, maternal level of education, household size and so on). I feel like the proliferation of such correlational studies with a low-explanatory power is already high enough in the literature on the effects of poverty on children. More importantly – and such criticism is not directed to the authors personally but towards a general trend – such superficial correlations often are interpreted in an appropriate (i.e., causal) way when it comes to deal with poverty. Even though I am personally convinced of the importance to provide further material and social support to low social background children, I do not believe that the present results strongly support this idea. We could also stress the fact that mothers should get psychological support. Or maybe mothers would not need psychological support if their children would get social and material support, who knows? I strongly believe that everyone – the scientific community, policy makers and citizens – would benefit more from greater effort put in studies design, data analysis strategies, as well as more challenging explanations in this literature.

Overall, I think that in its current form, the present manuscript does not offer a significant contribution to the literature on the effects of poverty on child mental health. To make a significant contribution, a greater deal of effort should be made to at least present more sophisticated data analyses (e.g., structural equation modeling) allowing richer interpretations and conclusions.

Minor comments

  • Lines 149-158: the way correlations are reported is confusing. For example, in model 1 “low income” positively correlates with “SDQ total score”, which indicates that the lower income and the greater children’ problematic behaviors. But the authors phrased it as a “negative association”. I understand that they meant that poverty has “negative” outcomes on children’ behavior but given the direction of the correlation and the way they present the factors, the correlation is actually positive. The others reported correlations in this section present the same issue.